# Early-Onset Diabetes in an Infant with a Novel Frameshift Mutation in LRBA

**DOI:** 10.3390/ijerph191711031

**Published:** 2022-09-03

**Authors:** Alessio Galati, Rosalia Muciaccia, Antonella Marucci, Rosa Di Paola, Claudia Menzaghi, Federica Ortolani, Alessandra Rutigliano, Arianna Rotondo, Rita Fischetto, Elvira Piccinno, Maurizio Delvecchio

**Affiliations:** 1Department of Pediatrics, Giovanni XXIII Children Hospital, Azienda Ospedaliero Universitaria Consorziale Policlinico, 70124 Bari, Italy; 2Research Unit of Diabetes and Endocrine Diseases, Fondazione IRCCS Casa Sollievo della Sofferenza, San Giovanni Rotondo, 71013 Foggia, Italy; 3Metabolic Disease and Genetics Unit, Giovanni XXIII Children’s Hospital, AOU Policlinico di Bari, Piazza G. Cesare 11, 70126 Bari, Italy

**Keywords:** early-onset diabetes, LRBA, primary immunodeficiency, hybrid closed-loop, bone marrow transplantation

## Abstract

We describe early-onset diabetes in a 6-month-old patient carrying an LRBA gene mutation. Mutations in this gene cause primary immunodeficiency with autoimmune disorders in infancy. At admission, he was in diabetic ketoacidosis, and treatment with fluid infusion rehydration and then i.v. insulin was required. He was discharged with a hybrid closed-loop system for insulin infusion and prevention of hypoglycemia (Minimed Medtronic 670G). He underwent a next-generation sequencing analysis for monogenic diabetes genes, which showed that he was compound heterozygous for two mutations in the LRBA gene. In the following months, he developed arthritis of hands and feet, chronic diarrhea, and growth failure. He underwent bone marrow transplantation with remission of diarrhea and arthritis, but not of diabetes and growth failure. The blood glucose control has always been at target (last HbA1c 6%) without any severe hypoglycemia. LRBA gene mutations are a very rare cause of autoimmune diabetes. This report describes the clinical course in a very young patient. The hybrid closed-loop system was safe and efficient in the management of blood glucose. This report describes the clinical course of diabetes in a patient with a novel LRBA gene mutation.

## 1. Introduction

Lipopolysaccarides-responsive beige-like anchor (LRBA) protein deficiency (OMIM #614700) is an autosomal recessive disorder caused by homozygous or heterozygous mutations in the LRBA gene [1]. The derived clinical syndrome has a wide spectrum of clinical manifestations, including immunodeficiency, autoimmune diseases, increased risk of infections, allergies and malignity, inflammatory bowel diseases, chronic pulmonary disorders, and type 1 diabetes mellitus [2,3,4,5,6,7].

Furthermore, LRBA variations are among the four genetic causes of very early onset autoimmune diabetes [8]. In fact, mutations in this gene should be suspected when a diagnosis of diabetes is made between 6 and 12 months of life in patients with autoimmune manifestations [8].

Most of these patients have negative pancreatic autoantibodies, suggesting that the autoimmune mechanism that causes diabetes may be different from that seen in type 1 diabetes. It could be that the autoantigens involved in the etiology of diabetes have not yet been identified, or that the autoimmune mechanism that causes pancreatic cell destruction is cell-mediated and not antibody-mediated [9,10,11].

Here, we describe a patient with early-onset diabetes who, on genetic analysis, turned out to be compound heterozygote for two mutations in the LRBA gene, a mutation inherited from the mother (c.2999del, p.Ser1000Tyrfs*2), until now never described, and one from the father (c.1963C>T, p.Arg655*), previously described [12,13]. We focused on blood glucose control and other comorbidities.

## 2. Case Report

The patient was born at full term (40 weeks + 2 days of gestational age, birth weight 3.95 kg). He was referred to the Metabolic Diseases Unit of Children’s Hospital in Bari at 6 months of life for diabetic chetoacidosis (glycemia 702 mg/dL, pH 7.24, glycated hemoglobin 5.9%). The patient had been showing inconsolable crying for 2 days. The fever appeared the day before admission (39.2 °C). Polyuria and polydipsia had occurred over the previous few days.

At admission, the patient was in fairly good general condition and presented Kussmaul respiration. The anterior fontanel was slightly depressed. Heart and abdomen were normal, and left hydrocele was detected. Weight was 7.6 kg (10–25 °C), and length was 66 cm (25–50 °C). The patient was promptly treated intravenously with saline solution of 0.90% *w*/*v* of NaCl (30 mL/h) for 2 h and then with 500 mL of 5% glucose solution with 20 mEq of NaCl and 20 mEq of KCl (25 mL/h) and insulin infusion (0.01 IU/kg/h) with further changes on the basis of blood glucose values. Subcutaneous insulin infusion with hybrid closed-loop system (Minimed670 G) was started in day 2 allowing stable blood glucose control.

Electrocardiogram, thyroid and abdominal ultrasound were normal. Biochemistry showed negative anti-GAD and anti-ZnT8 autoantibodies, while insulin and anti IA2 antibodies were positive (72.4 UA/mL, nv < 5; 12.2 UA/mL, nv < 5, respectively). The patient was discharged after 11 days, with an insulin requirement of 0.05 IU/kg/h from 8 p.m. to 8am and 0.125 IU/kg/h from 8 a.m. to 8 p.m. and with a bolus from 0.1 to 0.3 insulin units based on the blood glucose value detected at each feeding.

One week later, he was admitted to the hospital because of hyperglycemia (522 mg/dL) and fever. A jugular venous access was placed to manage fluids infusion and insulin therapy. Progressive glycemic compensation was reached, and defervescence was observed within 36 h.

Some days later, fever relapsed. A blood culture was performed, showing Staphylococcus Aureus and Klebsiella oxytoca infections. He was properly treated with antibiotic therapy with rapid defervescence. After removal of the venous access, a local edema progressively appeared, with overlying hot and hyperemic skin. A thrombosis of the right jugular vein was suspected and confirmed by an echo-color Doppler. A treatment with s.c. eparine (100 IU/kg twice a day) was started, with an improvement of the clinical condition. Seven days later, anemia (hemoglobin 8.2 g/dL) with positive direct Coombs test was found. Blood transfusion was then performed with hemoglobin normalization. Antibodies for toxoplasmosis, rubella, cytomegalovirus, herpes simplex, parvovirus B19, varicella-zoster virus and the anti-neutrophil membrane were negative, while the antinuclear antibodies (ANA) were positive.

At 9 months, he presented persistent diarrhea and growth retardation. The serological tests for celiac disease, RAST, fecal occult blood test, coproculture and research for gastrointestinal pathogens resulted negatives. Because of the persistence of diarrhea, at 13 months of age, he underwent an esophagogastroduodenoscopy and colonscopy. The former showed antral edema and hyperemia, nodular hyperplasia of the bulb, hypotrophy of the duodenal folds, and scalloping, suggestive of gastritis with nodular hyperplasia of the bulb. The findings suggested a possible inflammatory bowel disease (IBD). The latter showed that, at the level of the cecum, the colon presented erythematous mucosa, and the vascular reticulum appeared at times distorted without visible erosions. At the level of the proximal sigmoid, nodular hyperplasia of the mucosa was more marked than normal, and at the level of the rectum, there were minute aphthae, suggestive of IBD. The histological examination concluded that the picture was compatible with the diagnosis of celiac disease of grade B2according to Villanacci classification. The patient started Mesalazine (250 mg twice a day) with clinical benefits.

In the meanwhile, at 10 months, the patient presented swelling, pain and redness in the joints of the fingers and toes. After proper biochemical and clinical evaluation, polyarticular Juvenile Idiopathic Arthritis was diagnosed, and Prednisone (5 mg twice a day) was initiated.

In view of the very early onset of autoimmune diabetes, next-generation sequencing (NGS) analysis for monogenic diabetes genes was performed at 7 months of age. The proband resulted compound heterozygous for two mutations in the LRBA gene (NM_001364905.1), a deletion in exon 15 (c.2999del) p.Ser1000Tyrfs*2, never before described in other patients, and a nonsense mutation in exon 23 (c.1963C>T) p.Arg655*, previously described [12,13].

Both mutations were confirmed by Sanger sequencing in the proband and his parents. The mother was heterozygous for the first mutation, and the father for the second one. Of note, both mutations generate, if translated, two truncated proteins affecting the expression of the PH, BEACH and WD40 domains, and probably generate an LRBA deficiency. Figure 1 displays the protein domain maps and the identified mutation.

The patient underwent bone marrow transplant from his sister at the age of 16 months, who was heterozygous and turned to have an autoimmune thyroiditis with normal thyroid function. The transplantation was successful. Arthritis and diarrhea progressively reduced, he presented recurrent episodes of thrombocytopenia, which required immunoglobulins infusion. At last visit, at 24 months of age, the blood glucose control was excellent (HbA1c 6%, insulin requirement 0.6 IU/kg/day), and no severe hypoglycemia ever occurred. Growth was still stunted, height was 78 cm (<3° centile), and weight was 11.5 kg (10–25° centile).

## 3. Methods

After obtaining informed consent for genetic analyses, genomic DNA was extracted from EDTA whole blood using a QIAmp DNA Blood Kit (Qiagen Inc., Valencia, CA, USA).

The proband’s DNA was subjected to NGS analysis of the monogenic diabetes genes listed at https://www.operapadrepio.it/it/?option=com_k2&view=item&layout=item&id=5323 (accessed on 1 September 2021). DNA libraries were prepared by SureSelectXT-CCP17 (Agilent Technologies, Santa Clara, CA, USA), according to the manufacturer’s instruction, and sequenced on NextSeq-500 (Illumina Inc.). Regions of interest were covered >98% at a reading depth> 30x. References-based assembly was performed by Sequencher-v5.4.6 Software (GeneCodes, Ann Arbor, MI, USA) and variants assessed on a genomic scale and interpreted by Alamut Visual version 2.10 Interactive Biosoftware (SOFHiA GENETICS^TM^, Saint Sulpice, Switzerland). All identified mutations were validated by Sanger sequencing on a second PCR product.

## 4. Discussion

In this paper, we describe the clinical course of an infant with type 1 diabetes and immune dysregulation due to a LRBA gene mutation. LRBA protein is a member of the BEACH-WD40 protein family. It is involved in the immune response by actively reacting to the presence of lipopolysaccharide (LPS), but its defect appears also to be related to altered autophagic mechanism. This leads to the accumulation of damaged organelles and toxic substances in cells, and to a subsequent increase in the apoptosis mechanism. The increased apoptosis concerning B lymphocytes and cytotoxic T lymphocytes explains the augmented infectious risk of these subjects. Furthermore, it appears that patients affected by LRBA deficiency have a low level of at least two immunoglobulin isotypes (IgM, IgG, or IgA) [14,15].

On the other hand, the bigger apoptosis of T-reg cells results in the reduction in immune tolerance and therefore in autoimmune phenomena, as seen in other autoimmune diseases, such as systemic lupus erythematosus. In particular, LRBA appears to prevent lysosomal degradation of the CTLA-4 receptor. CTLA-4 is a powerful immune suppressor located in T lymphocytes with the task of blocking the co-stimulation of T cells, downregulating autoimmune processes. In fact, CTLA-4 mutations cause a disease called CHAI (CTLA-4 haploinsufficiency with autoimmune infiltration), which has almost overlapping symptoms with LRBA deficiency with autoantibodies, regulatory T (Treg) cell defects, autoimmune infiltration, and enteropathy (LATAIE). Abatacept is a drug mimicking the action of CTLA-4; therefore, it is capable of reducing autoimmune processes in patients with this mutation, and it is today one of the most important therapeutic options [16]. Another is the hematopoietic stem cell transplantation [12,17,18].

All mutations identified so far are associated with LRBA deficiency, which may cause a syndrome with a wide spectrum of clinical manifestations (i.e., autoimmunity, IPEX/like syndrome, immunodeficiency, infancy onset of type 1 diabetes and IBD) [2,3,4,5,6,7].

Interestingly, both of the proband’s LRBA mutations generate, if translated, two truncated proteins lacking the expression of the PH, BEACH and WD40 domains, and probably generate an LRBA deficiency. All the above domains, located in the C-terminal of LRBA protein, are highly conserved and participate in multiple cellular processes such as signal transduction, vescicular trafficking, and transcriptional regulation and are involved in the maintenance of intracellular stores of CTLA-4 by inhibiting its lysosomal degradation [3,12].

Our patient had diabetes as the first symptom, but the clinical picture soon became complicated with the onset of sepsis, anemia, and severe neutropenia. After a while, he also began to develop severe persistent diarrhea and growth failure, with the features of chronic IBD, and polyarticular arthritis of the toes and fingers with positive ANA. In all, they were manifestations of an ongoing autoimmune process.

Persistent diarrhea and growth retardation have been reported in patients with LRBA deficiency [19,20]. An association between LRBA deficiency and early-onset chronic erosive polyarthritis is described as well, and this prompted us to suggest that rheumatologists should suspect an LRBA mutation in the case of a patient with early-onset arthritis and other autoimmune phenomena [21,22].

No etiological treatment is available, but prophylactic antibacterial therapy and immunoglobulin replacement should be given, similarly to all primary immunodeficiencies. Despite all the currently available therapies for immunodeficiency, the control of the autoimmune mechanisms has been achieved only in patients treated with large doses of corticosteroids [3,23,24,25]. Treatment with sirolimus for inflammatory and autoimmune disorder has been reported as well [23]. More recently, a T-cell modulator, abatacept, has been considered as a targeted precision therapy [26].

Our patient presented a very early onset, 6 months, with several different autoimmune disorders before 1 year of age. Fortunately, we suspected a primary immune deficiency after the first few weeks of disease, and thus, a genetic diagnosis was available when he was 8 months old. This point allowed us to have frequent follow-up visits and, therefore, early diagnosis of all clinical manifestations. A systematic review reported an interquartile range at onset of 0.6–3.5 years, with a diagnostic delay in immunodeficiency of 1.0–9.7 years [27]. Our patient presented a very precocious early onset of clinical manifestation, and the molecular diagnosis was performed very early. We decided to perform hematopoietic stem cell transplantation, and 10 months after the procedure, we can say that most of the autoimmune mechanisms are properly controlled.

The clinical course of these patients is rarely reported in the literature, especially in such young patients. Diabetes mellitus has been reported in 24% of 109 patients with LRBA deficiency [27] and in some patients with LRBA deficiency, infantile-onset fulminant type 1 diabetes mellitus has also been described [3,25,28]. We focused our description on diabetes mellitus showing that blood glucose control was very good without any hypoglycemia, with an insulin requirement of 0.6 IU/kg/day. We did not observe any clinically significant change in insulin requirement and blood glucose control, and we think that these data will be interesting for clinicians, as they have not thus far been reported. After the bone marrow transplantation, arthritis and diarrhea progressively disappeared, while failure to thrive and diabetes mellitus persisted. We did not try to describe a genotype–phenotype correlation because reports from large cohorts concluded that a such correlation is not evident [27,29].

The novelty of this case is the identification of a novel frameshift mutation in the LRBA gene and the clinical admission investigation that may be worth reporting and useful for clinicians. Most previously published papers just summarize the clinical findings in their cohort, with scarce data about the timing of the other autoimmune disorders onset and their clinical course. The description of these rare patients provides supportive information for clinicians and researchers.

## 5. Conclusions

In conclusion, we describe two compound heterozygous LRBA mutations in an infant with early onset diabetes mellitus and autoimmune disorders, in whom treatment with bone marrow transplantation was successful. The patients carried two mutations in LRBA gene. The first, p.Arg655*, was previously reported in homozygosis [12] and recently described also in an Italian patient [13]. The second one, p.Ser1000Tyrfs*2, was here described for the first time. Mutations of the LRBA gene must always be suspected in the case of diabetes onset in the first year of life, in particular in the presence of other autoimmune and/or immunodeficiency manifestations.

## Figures and Tables

**Figure 1 ijerph-19-11031-f001:**
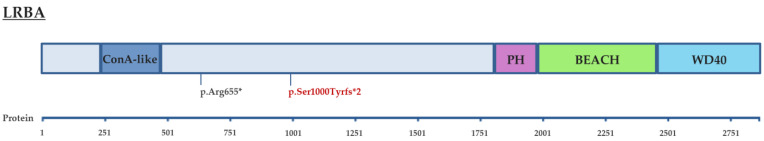
LRBA schematic protein domains map and location of the identified mutations. ConA-like: conconavalin A-like domain (186–371, 186 aa). PH: PH Pleckstrin homology domain (2073–2181, 109aa). BEACH: BEACH domain of Beige and Chediak-Higashi. WD40: WD40 repeat-like containing domain. Of the two identified mutations, the novel one is indicated in red text. The currently known disease-causing LRBA mutations are available on Human Genome Mutation Database at https://my.qiagendigitalinsights.com/bbp/view/hgmd/pro/start.php (accessed on 1 September 2021).

## Data Availability

Data are available on request in keeping with the current law.

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
