# Peer review of "Early-Onset Diabetes in an Infant with a Novel Frameshift Mutation in LRBA"

_ijerph, 2022, doi:10.3390/ijerph191711031_

Round 1
Reviewer 1 Report
The manuscript of “Early onset diabetes in an infant with LRBA mutation” by Galati et al. described the medical care management in an infant with a very early onset autoimmune diabetes who carried a compound heterozygous mutations in the LRBA gene. The recessive null mutations in LRBA are associated with neonatal diabetes diagnosed <1 year, which had been not rarely reported, e.g.Johnson. et al. (Diabetes 2017), and Schreiner. et al (JCEM2016). The novelty of this study is to identify a novel frameshift mutation in LRBA and clinical admission investigation, that may be worthy to report.
Some comments:
1. Since it was known recessive compound null mutations in LRBA result in early-onset diabetes and other immunodeficiency manifestations, any conclusion about “rare form of diabetes”, or “expands the spectrum” should be avoided. This study is consistent with previous studies. It will be misleading for readers not to describe the previous studies. For this point, the tile somehow needs to be changed into a more special one.
2. There is a nomenclature error in one mutation c.2999_300delC, it should be c.2999del which was only one base deletion and HGVS nomenclature required the RefSeq transcript no. when you describe a variant. NM_001364905.1 is a MANE transcript that may be used for your variant.
3. The English writing is fluent and smooth. The main problem is that we could not figure out the difference between other studies or whether they are very similar. The description of “clinical case” is like a regular blog of a clinician. The paragraph was for a clinical episode but not for understanding the novel findings. It is necessary to highlight what you want to pick up.
4. The discussion on the correlation between genotype and phenotype in this study is not sufficient. Comparison to the previously reported cases with similar variants is necessary.
5. Is the outcome of bone marrow transplant better than other patients with LRBA mutation?
Author Response
We would like to thank the reviewer for the his/her comments. We amended the manuscript in keeping with the suggestions. Please find below the point-by-point responses:
- The paper was been amended in keeping with this comment. In particular, the last sentence of the abstract was replaced by ”This report describes the clinical course of diabetes in patient with a novel LRBA gene mutations”, which seems to be more appropriate
- Thank you for this observation. The nomenclature has been updated
- We thank the Reviewer for his/her comment. As he/she highlighted, the novelty of our study resides in the description of a novel frameshift mutation in LRBA together with a detailed clinical picture, that we believe is worthy to report. We recognize this is the main difference between our study and other clinical reports, thus all this is now stated in the Conclusion section of the revised manuscript.
- Thank you for this comment. Actually, a genotype-phenotype correlation is not evident on the basis of data from literature. This point was added in the manuscript and it accounts for the lack of a genotype-phenotype correlation evaluation.
- No, up to date the outcome is not different from the other patients.
Reviewer 2 Report
This study mainly describes the clinical case.
It is recommended to submit to a clinical journal rather than this journal.
Author Response
Thank you for this comment. We chose to submit our case report to this Special Issue because it is focuses on “Screening, Prevention, Diagnosis and Therapy”. We think that our paper is suitable for this Special Issue, which aims to collect clinical reports, as we describe an early genetic diagnosis (Diagnosis) which allow clinicians to suspect diseases which are uncommon in so young patients and to avoid delay in treatment (Secondary Prevention). Finally, we focus on the treatment of the disease (Therapy) to provide useful information for other clinicians.
Reviewer 3 Report
This articte is suitable to accept for this unique journal. However, it needs some corrections.
1. Line 62 th...''...physiological solution..'' What is your meaning by saying pyhsiological solution? Could you make it more clear?
2. Line 63 th...''....%5 glucose solution with electrolytes ...'' which kind of electrolytes you mean?
3. Line 82 th..''...sc eparine..'' eparine? you mean heparine? which dosage per kg instead of total dosage of the day
Kind Regards.
Author Response
- The sentence has been modified. We hope it is more clear now.
- The sentence has been modified. We hope it is more clear now.
- Kind Regards. We added the dosage of heparine per kg.
Reviewer 4 Report
In the MS “Early onset diabetes in an infant with LRBA mutation” there is a detailed report of symptoms, diagnosis, treatment and follow-up of very young diabetics with mutations in the LRBA gene. Using NGS authors detected two mutations in LRBA gene in this patient. One mutation (p.Arg655*) was reported previously, while other mutation (p.Ser1000Tyrfs*2) authors described for the first time in this report. Manuscript is submitted as an Original article, however, it is written in a Case report form.
1. Could authors show schematic structure of LRBA gene with denoted mutations?
2. In line 99: “The histological examination concluded that the picture was compatible with the diagnosis of celiac disease of grade B2 according to Villanacci classification.”Could authors show histological sections?
3. Sentence in the Introduction “In fact, mutations in this gene should be suspected when diagnosis of diabetes is made between 6 and 12 months of life, in patients with autoimmune manifestations” requires Reference.
4. Sentence in the Discussion “All mutations identified so far, are associated with LRBA deficiency that may cause a syndrome with a wide spectrum of clinical manifestations (i.e. autoimmunity, IPEX/like 161 syndrome, immunodeficiency, infancy onset of type 1 diabetes and IBD).” lines 160-162 requires References.
5. In line 57, “39.2 C°” should be corrected.
6. Full stops and commas are missing in some References in Reference list.
7. In line 148, the abbreviation T-reg should be explained, since it is mentioned here for the first time.
Author Response
- We thank the Reviewer for his/her helpful suggestion. A picture showing the LRBA schematic structure with the 2 denoted mutations has been added in the revised version of our manuscript.
- No, unfortunately the figure of the histological section is not available.
- The reference is the same of the previous sentence, ref. 8. It has been added in the manuscript.
- The reference has been added.
- Done.
Round 2
Reviewer 2 Report
It is sufficient to be published in a journal as a case report.